# Towards On-Device Dehydration Monitoring Using Machine Learning from Wearable Device’s Data

**DOI:** 10.3390/s22051887

**Published:** 2022-02-28

**Authors:** Farida Sabry, Tamer Eltaras, Wadha Labda, Fatima Hamza, Khawla Alzoubi, Qutaibah Malluhi

**Affiliations:** 1Computer Science and Engineering Department, Faculty of Engineering, Qatar University, Doha 2713, Qatar; tamer.taras@qu.edu.qa (T.E.); wadha.lebda@qu.edu.qa (W.L.); fatmaa@qu.edu.qa (F.H.); qmalluhi@qu.edu.qa (Q.M.); 2Engineering Technology Department, Community College of Qatar, Doha 7344, Qatar; khawla.alzoubi@ccq.edu.qa

**Keywords:** on-device, dehydration detection, wearable devices, hydration monitoring, machine learning, skin response, electro-dermal activity, photoplethysmography

## Abstract

With the ongoing advances in sensor technology and miniaturization of electronic chips, more applications are researched and developed for wearable devices. Hydration monitoring is among the problems that have been recently researched. Athletes, battlefield soldiers, workers in extreme weather conditions, people with adipsia who have no sensation of thirst, and elderly people who lost their ability to talk are among the main target users for this application. In this paper, we address the use of machine learning for hydration monitoring using data from wearable sensors: accelerometer, magnetometer, gyroscope, galvanic skin response sensor, photoplethysmography sensor, temperature, and barometric pressure sensor. These data, together with new features constructed to reflect the activity level, were integrated with personal features to predict the last drinking time of a person and alert the user when it exceeds a certain threshold. The results of applying different models are compared for model selection for on-device deployment optimization. The extra trees model achieved the least error for predicting unseen data; random forest came next with less training time, then the deep neural network with a small model size, which is preferred for wearable devices with limited memory. Embedded on-device testing is still needed to emphasize the results and test for power consumption.

## 1. Introduction

Following the tremendous technological advances in the design of systems on a chip (SoCs), the development and use of wearable devices have remarkably achieved high growth rates in the last few years. Wearable devices for different biosignals’ acquisition have been researched [1], and challenges to and opportunities for wearable device design, biosignal analysis, and interpretation have been discussed [2]. These advances in the research and industry of wearable devices drive the research and development efforts to use wearable devices to monitor, analyze, and learn from different sensors’ data for different people’s healthcare applications.

One of the main health issues that is usually ignored by many people, and which can lead to serious health consequences if ignored, is dehydration. While dehydration is more common in countries with very hot and humid weather where summer temperatures can reach 50 °C, it can also be seen in cold countries [3]. The hydration level is greatly affected by many factors: water intake, body temperature, thirst drive, the health conditions of a person, the frequency of urination, etc. Some of the symptoms associated with dehydration include dry mucous membranes, dry axilla, tachycardia, poor skin turgor, low blood pressure, urine color changes, and dizziness in some cases [4]. While persons normally feel thirst when they are going into dehydration to alert them to drink enough water to feel better, this is not the case with elderly people who cannot communicate their feeling of thirst and for people with impaired thirst drive, a condition known as adipsia. Additionally, people whose work requires laborious non-stop effort in very hot or very cold weather, such as soldiers and construction workers, may miss drinking regularly. These cases have great need for dehydration monitoring.

To check the dehydration level of a subject, doctors often clinically use invasive methods (e.g., blood tests) or offline methods (e.g., urine tests) to obtain parameters such as plasma osmolality (PO), total body water (TBW), bioelectrical impedance (BIA), urine characteristics (color, specific gravity), salivary osmolality and flow rate [4], as well as measuring blood pressure and the pulse rate and observing the skin and eyes of the subject. Using wearable devices, without the need for these invasive tests and doctor’s checkups, for monitoring dehydration and alerting the user before he/she goes into dehydration, can be of great help. Using a smartwatch or device to monitor dehydration can be particularly beneficial for the geriatric population who have problems in communicating their need for water and for the other above-mentioned situations.

One approach is vision-based devices that use monitoring cameras to monitor the subject and record the eating and drinking moments [5]. However, this approach faces some challenges as the device needs to be on all the time to record and detect these moments, which is not very practical for limited-power wearable devices and for performing daily activities freely. It also does not consider the body signals and the different needs for water and food intake. Wearable devices that can read some of the body signals related to the aforementioned symptoms of dehydration and then decide based on machine learning techniques the hydration state of the subject seem to be a promising approach [4,6,7]. With the growth of using smart devices and improved hardware technology, there is a growing interest in performing machine model training on the device [8]. This on-device-trained model may be used in an online active learning setting for inference or it can be used to update a global model in a federated setting [9]. In this research, we approached the dehydration detection problem using machine learning as per the pipeline in Figure 1. With the idea of on-device learning in mind for privacy requirements and for ease of use with the resource-constrained characteristics of wearable devices, we tested different models and learning approaches. We conducted comparison between different machine learning algorithms and testing in a transfer learning setting for personalization.

The contributions of this research to the field of hydration monitoring can be summarized as follows:
We propose modeling the dehydration problem as a regression problem to predict the last drinking time using features extracted and constructed from multiple wearable sensors (accelerometer, magnetometer, gyroscope, galvanic skin response (GSR) sensor, photoplethysmography (PPG) sensor, temperature, and barometric pressure sensor);For this purpose, we recorded a total of 3386 min for these sensors for 11 subjects under fasting and non-fasting conditions;We compared different machine learning models for this task according to four metrics that evaluated the accuracy of the model, training time, and model size and for on-device personalization optimization through the transfer learning experiment;A comparison to the state-of-art research for this task was performed. Challenges and limitations were discussed, and further research directions were highlighted.


The rest of the paper is organized as follows. In Section 2, the related work for hydration monitoring using wearable devices is reviewed and the necessary background for this field is presented. Our methodology for applying machine learning to detect dehydration is presented in Section 3. The dataset description, specific details about the different body signals, data preprocessing, and descriptive statistics for the dataset are provided in Section 4. In Section 5, the results of applying different machine learning models to the data are presented. Section 6 discusses the results and compares the study to previous studies for dehydration detection. The conclusion and summary of the results are presented in Section 7.

## 2. Background and Related Work

There is ongoing progress in the development of biochemical sensors that can measure the concentration of different electrolytes in sweat and hence determine the hydration state of the human body [10]. Side by side with the great ongoing research to develop different types of sensors [11,12], machine learning research is going on to learn from body signals that can be collected from different sensors to detect dehydration based on the effect of cognitive stress triggered by dehydration on the autonomic reactions of the body [7,13,14]. In this section, a short overview is presented for the previous studies in dehydration detection based on machine learning. Performing machine learning on-device is also briefly reviewed to have the advantages, as well as the challenges in mind when modeling the machine learning experiments for wearable devices and how transfer learning was employed in these tasks.

### 2.1. Dehydration Detection

As discussed in the Introduction, dehydration detection has been researched using either invasive techniques in clinical setting or in a non-invasive way using machine learning techniques to learn from the features of some body signals. There are relatively few studies that have worked in this area, which we briefly review for the used features, techniques, and evaluation metrics.

Features extracted from GSR sensors or sometimes called electrodermal activity (EDA) sensors have been used in most of the studies [4,6,7,15]. They used statistical features such as minimum, mean, variance, entropy, standard deviation, percentile, median, mode, and kurtosis of skin resistance, as in [4], for the ease of extraction, but it is hard to interpret how these statistical features affect the dehydration status. The authors in [7] combined pulse rate variability parameters extracted from PPG signals with EDA for mild dehydration identification. In [15], the authors integrated with the EDA features the output labels of the activity recognition model with four postural activities (walking, sitting, standing, and lying) and two hand gestures (drinking water and not drinking water) as features for the dehydration classification model. Differently, the authors in [14] investigated the use of heart rate variability (HRV) parameters: RR-interval of the ECG signal, standard deviation of RR-interval (SDRR), and root mean sum of squares of differences between adjacent RR-intervals (RMSSD) in the ECG signal as features. They did not obtain the ECG signal from a wearable device under free moving conditions, but rather from the Cardiosoft device in a clinical setting.

Many machine learning techniques have been tested in these studies. Linear discriminant analysis (LDA), quadratic discriminant analysis (QDA), logistic regression (LR), support vector machine (SVM), Gaussian kernel, k-nearest neighbor (KNN), decision trees, and ensemble of KNN were used in [7] with ensemble KNN achieving the best classification accuracy of 91.2%. In [6], they tested LR, SVM, decision trees (DTs), KNN, LDA, and Gaussian Naive Bayes (NB), and KNN achieved the best classification accuracy of 87.78%. SVM and K-means were used in [14], and SVM was better with only a 60% classification accuracy. In [15], random forest (RF), DT, NB, BayesNet (BN), and multilayer perceptron (MLP) classifiers were tested with the maximum accuracy for decision trees with 93% accuracy. LR, RF, KNN, NB, DT, LDA, the AdaBoost classifier (ABC), and QDA were tested with a maximum accuracy of around 91% for random forest and decision trees.

All the aforementioned studies approached the problem as a classification problem, either as binary classification with hydrated/dehydrated labels or with more fuzzy levels. The problem with this approach is two fold, the difficulty of accurate reliable labeling of the data and neglecting the personal differences between subjects. The classification models in all the studies were evaluated according to the classification accuracy; some studies included other classification metrics such as sensitivity and specificity in [7] and precision, recall, and the F1-score in [4]. Power consumption analysis was performed only in [15] based on battery level consumption for running the algorithms on data streamed for 5 h on an Android platform. None of these studies performed a comparison related to on-device deployment in terms of training time, inference time, and model size to optimize on-device deployment for wearable devices. We elaborate more on this point, the used features, and the dataset size in the Discussion Section 6 when comparing them to our study.

### 2.2. On-Device Machine Learning

Deploying the machine learning model on the device has the advantages of keeping the data private and decreasing the latency for the prediction/classification as there is no need to transmit large amounts of data from the device to the cloud. Specifically, for healthcare applications, having the patient’s data and the machine learning model on the device is safer from a privacy-preserving perspective. Low latency and real-time feedback are also required for many healthcare applications that require immediate alert for the users or their care givers such as fall detection, stress detection, etc. On the other side, the main disadvantages for on-device computing include the limited device computing power, storage, and battery life. These obstacles are being addressed in the recently growing field pf tinyML [16] through model compression and quantization techniques. The authors in [17] proposed tiny transfer learning (TinyTL) to freeze the weights of the feature extractor while only learning the biases, thus not requiring storing the intermediate activations, which is the major memory bottleneck for on-device learning. They showed that TinyTL can reduce the training memory cost by an order of magnitude (up to 13.3×) without sacrificing accuracy much. Transfer learning (TL) means storing knowledge gained while solving one problem and applying it to another related problem. Transfer learning is commonly seen as a technique that involves using a pre-trained model for one task and retraining a part of its layers (typically the last ones) to solve another task.

The authors in [18] developed a personalized, cross-domain 1D CNN by utilizing transfer learning from an initial base model trained using data from 20 participants completing a controlled stressor experiment. By utilizing physiological sensors (HR, HRV, EDA) embedded within edge computing interfaces that additionally contained a labeling technique, it was possible to collect a small real-world personal dataset that could be used for on-device transfer learning to improve model personalization and cross-domain performance. In a similar fashion, the authors in [19] studied applying Bayesian active learning for embedded/wearable deep learning to the HAR and fall detection scenarios. This facilitated on-device incremental learning (model updating) for seamless physical activity monitoring and fall detection over time.

In short, transfer learning can be employed in tasks when not many data are available for training, as well as for model personalization when the accuracy of the model is highly dependent on the subject under test. By performing this on-device [17], the privacy preservation of user’s data is ensured as well.

In this paper, we first investigated the application of machine learning techniques for dehydration detection using body signals and features extracted from them. A comparison of different models was then performed based both on the average error across subjects, the accuracy of the transferred learned model from subjects to produce a personalized model for another subject, and the size of the model to be deployed on-device.

## 3. Machine Learning for Dehydration Monitoring

This research paper proposes using machine learning to learn from sensors’ data recorded from different subjects and predict the last drinking time. The trained machine learning model was then to be deployed on-device to guarantee the privacy of data and to personalize the model according to the online learning from new recorded data, as mentioned in the last section. The pipeline for the machine learning stages is shown in Figure 1. A similar technique was used in [18] for personalized stress modeling.

All the dehydration monitoring/detection research work referenced in the last section [4,6,7,14,15] approached the problem as a classification problem. They learned from some body signals and statistics derived from them with labels for the subject’s hydration level (either hydrated or dehydrated or with more fuzzy levels of well-hydrated, hydrated, dehydrated, and very dehydrated). An assumption was made to consider 10 h of fasting as dehydration in [4].

Differently, in this paper, we approached the dehydration monitoring problem as a regression problem. The problem was represented as a prediction problem for the number of hours since the subject last drank water. Ten hours of fasting does not necessarily lead to dehydration. Many Muslims usually fast, sometimes for 16 h a day or more, without any water or food intake and without suffering from dehydration or related symptoms. Additionally, dehydration differs greatly from one person to another according to personal characteristics and other health conditions. With IRB approval, subjects were asked to voluntarily participate and record data normally at different times, some while they had their last water intake within an hour, others during Muslim fasting days (e.g., in the fasting month of Ramadan), and some during intermittent fasting hours after waking up and before any water intake. Sensors’ data, as well as some subject characteristic data were collected as discussed in the next section. Following the data collection process, preprocessing of data, feature extraction, and analysis took place in the preprocessing step in the pipeline in Figure 1. Thereafter, training machine learning models took place using the collected preprocessed data. The ML trained model was used for on-device deployment to learn from the personal data in a private and secure way, where the user’s data did not leave the device.

On-device machine learning can take place in three different settings. Firstly, this can occur through having a pretrained model with the whole set of features deployed on the device, which can be used directly for inference, and the model’s parameters are updated through online learning with confirmations from the user for the predicted output. The second setting is a transfer learning setting where a pretrained model is deployed and either the same set of features is fed or an extra set of features is extracted to train a new model that is different in parameters and/or structure [20]. Finally, a federated learning setting can be employed where each device learns its own model and the devices collaboratively learn a global model either through an orchestrating server or in a decentralized way between them [9]. In this research work, we simulated and evaluated the first two settings where the user’s data and the updated model parameters do not leave the device and the same set of features was used in the model evaluation and on the device.

## 4. Dataset, Preprocessing, and Feature Extraction

For data collection, we used the Shimmer Galvanic Skin Response (GSR) unit (http://www.shimmersensing.com/products/gsr-optical-pulse-development-kit (accessed on 18 February 2022)). The device was calibrated according to the procedure provided by the Shimmer Calibration User Manual. Subjects who were likely fasting during Ramadan or were voluntary fasting wore the device in two scenarios during fasting hours in the month of Ramadan or in an intermittent fasting scenario with no food or water intake after waking up in the morning for a few hours, recording the last time for water and food intake. No restrictions on movement or the time of wearing the device were applied to represent a real-world scenario. Some data were collected while a subject was driving a car or walking, but most of the data were recorded while the subjects were sitting at their desks. The Shimmer GSR unit was worn on the subject’s wrist with two electrodes attached to the index and middle finger of the left hand and the PPG clip on the left ear lobe. All data collection was subject to Qatar University Institutional Review Board (IRB) approval procedures covered by the IRB approval: QU-IRB 1538-EA/21. The raw dataset has been made available at Zenodo (https://zenodo.org/record/6299964).

### 4.1. Sensors’ Data Preprocessing

Data collected from the Shimmer3 GSR unit included different sensors’ data: accelerometer, magnetometer, gyroscope, GSR sensor, PPG sensor, temperature, and pressure sensors (https://shimmersensing.com/support/wireless-sensor-networks-documentation/ (accessed on 18 February 2022)).

#### 4.1.1. Accelerometer Data

Accelerometer data were measured from three channels for *x*, *y*, *z* acceleration with a magnitude sample, as shown in Figure 2. The specification of the accelerometer used in Shimmer is added in Table A1 in Appendix A. The magnitude of accelerometer data in (*x*, *y*, *z*) was calculated as in Equation (Equation 1) from the calibrated sensor values. The Shimmer’s accelerometer operates at a 512 Hz sampling frequency by default with a maximum range of +/−16; however, the sensor occasionally gives readings beyond these limits, and it is not recommended to rely on out-of-range measurements. We clipped readings exceeding the range to the maximum in the same direction. The sampling frequency could be reduced so much, as in the dehydration task, we were not concerned with the fine details of the signals, but rather with changes over longer time intervals. However, we used the default sampling frequency so that the dataset can be useful for other tasks, and we also downsampled the data at 1 min intervals.

As a step of feature engineering, we calculated the accumulative change in accelerometer magnitude as in Equation (Equation 2) as a representative feature for the total change in velocity from the start of streaming users’ signals. The change in accelerometer magnitude between two consequent time steps is known as the motion jerk. It was used in [21] for robust activity recognition, in [22] for quantifying bi-manual arm use, and in [23] for the classification of sleep–awake states. Though this calculation is considered an approximation of the total change in velocity, in this application, the exact value for the change at each time step was not important as we did not calculate the distance or time from it. In those cases, calculating Euler angles to predict the motion performed by a person would be more accurate [24]. It can reflect the effort and state of activity of the subject together with the accumulative change of the magnetometer and gyroscope magnitudes.
(1)$$\left|ACC\right|=\sqrt{\left({acc_{x_{t}}}^{2}+{acc_{y_{t}}}^{2}+{acc_{z_{t}}}^{2}\right)}$$
(2)$$cumAcc=\sum_{j=1}^{t-1}{\left|ACC\right|}_{j+1}-{\left|ACC\right|}_{j}$$


#### 4.1.2. Magnetometer Data

The magnetometer data were measured from three channels for the *x*, *y*, *z* magnetometer at a 512 Hz sampling frequency with a magnitude sample, as shown in Figure 2. The calculation of the magnitude of the magnetometer data in (*x*, *y*, *z*) was performed as in Equation (Equation 3) from the calibrated sensor values. The specification of the magnetometer used in Shimmer is added in Table A2 in Appendix A. Another feature representing the cumulative sum of the magnitude of magnetometer readings was calculated from the data as in Equation (Equation 4).
(3)$$\left|MAG\right|=\sqrt{\left({mag_{x_{t}}}^{2}+{mag_{y_{t}}}^{2}+{mag_{z_{t}}}^{2}\right)}$$
(4)$$cumMag=\sum_{j=1}^{t-1}{\left|MAG\right|}_{j+1}-{\left|MAG\right|}_{j}$$


#### 4.1.3. Gyroscope Data

The gyroscope data were measured from three channels for the *x*, *y*, *z* gyroscope with the magnitude, as shown in Figure 2. The specification of the gyroscope used in Shimmer is added in Table A3 in Appendix A. The calculation of the magnitude of the gyroscope data in (*x*, *y*, *z*) was performed as in Equation (Equation 5) from the calibrated sensor values. Another feature representing the cumulative sum of the magnitude of gyroscope readings was calculated from the data to reflect the movement and activity level of the subject as in Equation (Equation 6). The gyroscope data were the signals most affected by the motion of the subject, which is why they were used for adaptive filtering of the PPG signal, as discussed later.
(5)$$\left|GYRO\right|=\sqrt{\left({gyro_{x_{t}}}^{2}+{gyro_{y_{t}}}^{2}+{gyro_{z_{t}}}^{2}\right)}$$
(6)$$cumGyro=\sum_{j=1}^{t-1}{\left|GYRO\right|}_{j+1}-{\left|GYRO\right|}_{j}$$


#### 4.1.4. GSR Data

This represents the galvanic skin response, which includes both the phasic and tonic level components, as shown in Figure 3. It is sometimes referred to as electrodermal activity (EDA). EDA has been used for many applications such as stress detection [25] and other mental health tasks [26]. Additionally, it has been used for other pathophysiological applications such as fatigue, pain, and sleepiness assessment, as well as the diagnosis of epilepsy and neuropathies [27]. The specification of the GSR sensor used in Shimmer is added in Table A4 in Appendix A. From the raw data recorded by the GSR sensor, there were two negatively correlated features: GSR conductance measured in μS and GSR resistance measured in kOhms, since they are reciprocals of one another. As mentioned by [27], the phasic changes of the EDA signal hold information about rapid changes [27] that correspond to responses to short events.

#### 4.1.5. PPG Data

The PPG data can be captured using an optical photoplethysmography sensor (PPG) that measures changes in the arterial translucency using a light emitter and detector. Arteries absorb less light when the heart pumps blood to the different body parts, and so, less light is received by the detector. This change in the amount of light absorbed and reflected is used to measure changes in blood volume in the arteries and capillaries, which can be used to estimate heart rate and heart rate variation. The specification of the PPG sensor used in Shimmer is added in Table A5 in Appendix A.

The PPG signal is greatly affected by motion artifacts, as shown in Figure 4, and there has been much research performed in the area of artifact removal [28]. The gyroscope has been found to be the most sensitive sensor to motion [28], so we used the mean of the cumulative change of the gyroscope for adaptive filtering to preprocess the PPG data using a low-pass filter with different cutoff frequencies for high-motion and low-motion periods. The output of the adaptive filter was then followed by a smoothing filter to remove unnecessary details so that it could be processed after that to extract features such as the inter-beat interval (IBI), beats per minute (bpm), breathing rate (BR), and the root mean square of successive differences between normal heartbeats (RRMSSD). These features were extracted from the cleaned PPG signal using the HeartPy algorithm [29]. When the heart rate estimation algorithm failed, the data were imputed with the last value in the previous interval. The preprocessing steps for the PPG signal are summarized in Figure 5.

#### 4.1.6. Temperature and Pressure Data

Shimmer3 uses the BMP280 barometric pressure and temperature sensor for measuring barometric pressure with a ±1 hPa absolute accuracy and temperature with a ±1.0 °C accuracy. Atmospheric temperature and pressure can affect the hydration state of a person, especially in hot weather and very cold weather, as previously mentioned [3]. We added the calibrated outputs of the BMP280 sensor downsampled at 1 min intervals, although most of the data collected in this study were for subjects in room temperature conditions.

### 4.2. Descriptive Statistics

The data were collected from 11 healthy subjects (9 females and 2 males with a mean age of 30, a mean height of 158 cmand a mean weight of 62 kg) with no underlying known health problems in a desk working conditions scenario. Though we approached the problem as a regression problem, we split the data into fasting and non-fasting samples. Non-fasting samples included data collected when the subject was hydrated and drank water within the last hour. Fasting samples contributed to the data collected from fasting subjects as mentioned before. The data collection was challenging due to COVID-19 restrictions. A total of 3386 min (56.4 h) was recorded using the Shimmer 3 GSR Development Kit. A summary of the duration of samples collected from each subject and some descriptive statistics are presented in Table 1.

### 4.3. Features

Based on the raw sensors’ data features (48 features) recorded by the ShimmerCapture software, we selected some of the raw calibrated version of the signals in addition to some extracted features, as mentioned in the last subsection, and some personal characteristic features to have a feature vector with a total length of 19. A list of the used features (we refer to it as FEAT1 in the rest of the paper) is listed in Table 2 together with the mean value for each. The data were downsampled at 1 min intervals with a sampling rate of 0.017 Hz, aggregating the sensors’ data and other features calculated from them with the mean values. Age, height, weight, and gender were used as personal characteristic features.

To analyze these features (FEAT1) and understand the inter-dependencies among them, basic correlation analysis was performed. Figure 6 shows the existence of high positive correlation between the variables in each of the following sets: {PPG_A13_CAL, Accel_mag, Mag_mag, Gyro_mag}, {cumAccel, cumMag, cumGyro}, and {height, weight}. It also shows a negative correlation between {bpm and ibi} since the increase in the inter-beat interval means less beats per minutes. To assess multicollinearity and its severity level, the variance inflation factor (VIF) was calculated—the values are shown in the features’ table in Table 2 with some values above 5 and a few above 10. The importance of the features for the predictability of the target output (sincelastDrinking) using the random forest algorithm was performed, and the result is shown in Figure 7.

According to these results, eliminating some features and combining some features were performed. Height and weight were replaced by the body mass index (BMI). The accelerometer, magnetometer, and gyroscope features of the magnitudes and cumulative sum were replaced by a summation of the ratios of the cumulative sum of the magnitude to the magnitude feature of each. The ibi feature was eliminated as bpm was linearly calculated from ibi, so including one of them was sufficient. The new feature set had 12 features, and it is referred to as FEAT2. We did this in favor of overcoming the multicollinearity, as well as to follow the Occam’s razor principle, which means in this case that fewer features were preferred as long as they do not affect the model’s accuracy, as will be shown in the next section. Using fewer features means simpler models, which require less memory for storing features and parameters and less training time, which were desirable for the case at hand of limited resources in wearable devices. A comparison of the performance using the two features sets is presented in the next section.

## 5. Results

In the first set of experiments, comparing the different machine learning algorithms was performed with the whole dataset (FEAT1) of the 11 subjects with a 70/30 training/testing split with the baseline of predicting with the median value. The averaged results over 10 runs are shown in Table 3 and Figure 8. It is clear from the results that linear models such as linear regression, its regularized versions (ridge, lasso, ElasticNet), support vector regression (SVR), and the simple linear artificial neural network (ANN) did not perform as well as the non-linear models. Non-linear models, such as tree-based models and deep neural networks (DNNs), are more capable of capturing the non-linear relationship between the input variables and the target variable. This comes at the expense of more training time and more complex models, which requires more storage.

Using FEAT2 was found to achieve less error, as shown in Table 4. As can be seen in Table 3 and Table 4, extra trees, random forest, and DNN achieved the least mean absolute error and least root-mean-squared error in both experiments, and they showed stable performance. Random forest was the best of them in terms of the training time, and DNN was the best in terms of the model size. The model size that was computed here was the size of the model trained on an Intel processor and saved to disk, not the size of the model in memory for embedded deployment, but still, this gave an indication of the complexity of the model. For production, these models can be converted to a lite version, which can cause minimal loss in accuracy. Many optimization techniques were used in this conversion process to perform transfer learning on-device through quantization, pruning, hashing, data compression, etc. [8].

To evaluate how these three models (extra trees, random forest, and DNN) performed in a transfer learning setting with a fixed structure and features, another experiment was performed with the models trained with data for all subjects except one and retrained with part of the data as a training set to tune the parameters of the mode to personalize it, which can represent the labeled data that the user or his/her caregiver provide as the input or correction to the predicted output. Table 5 shows the comparison of the three techniques for the mean performance across all subjects chosen as test subjects. The results are for both testing with a small set, which represented 0.3 of the subject’s dataset, which means retraining the model with 0.7 of the data, and with testing with 0.7 of the data, which means training with 0.3 of them. The results showed that the average MAE error increased slightly more than the average shown in the previous experiment in Table 4, which is normal because all the samples of the new subject were new to the model. The maximum increase in error was when using the DNN model, where the average MAE in prediction increased by about 2 h. However, for random forest and extra trees, the maximum increase in error did not exceed 0.4 (24 min) when retraining the model with only 0.3 of the subject’s data and testing with the rest. The mean training time of random forest and mean model size on disk were the least, which means less computational and memory requirements. Both the MAE and RMSE were slightly better for extra trees than random forest, but with the added complexity of the model.

The random forest model was found to achieve good comparable accuracy with respect to the prediction error, as well as the least training time, which is preferred for wearable devices with limited computational resources. It inherits interpretability and explainability from decision trees and is recommended to be used in different healthcare applications. With respect to the model size, more optimization can be performed by applying TinyTL techniques to reduce the training memory footprint by an order of magnitude for efficient on-device learning through quantization and other strategies [17] and testing for the actual model size in memory. More experiments and benchmarking [30] to assess the power consumption, latency, and accuracy of the model and runtime on TinyML hardware need to be performed to gain more confidence in the results.

## 6. Discussion

In this section, the results of this research are compared to the related papers in the literature reviewed in Section 2, which approached the same task of hydration/dehydration monitoring and detection, as shown in Table 6.

The first point of comparison is how the problem was modeled. All the other studies modeled it as classification into two, three, or four classes, as summarized in Table 6. The problem with this approach is how the labels of the samples are assigned and the assumptions made. In [4,6], the authors assumed a dehydrated state after fasting for 10 h without any fluid or food consumption. In [15], they followed a protocol where the fasting subject labeled the data at first as well-hydrated and then switched to hydrated when he/she felt thirsty for the first time and then to dehydrated for the second time and very dehydrated when feeling extremely thirsty, such that his/her tongue went dry. This protocol can be seen as difficult to follow, as well as many fasting Muslims as those recruited in their study may not reach the dehydration state even after fasting for 16 h, as it depends on the effort exerted during the day, the water and food intake in the previous day, as well as the health conditions of the subject. The authors in [7] were more conservative and used the label “mild dehydrated” for data on the day of fluid restriction. In [14], the study was for athletes and the effort exerted during exercise, so they used three classes before exercise, post-exercise at 37 min, and after hydration. In this study, the problem was modeled as a regression problem for the number of hours since the subject last drank water. This approach has the advantage of the ease of collecting data where only the last time of drinking is added for a sample of recording the data, and the target variable was added programmatically according to the timestamp. Additionally, predicting the last drinking time provides the flexibility of adjusting the alerting threshold on the wearable device based on the health conditions, needs, and activity of the user. The alerted user/caregiver can then choose to input the correct drinking time if the prediction was not accurate for his/her situation to enable a closed-loop solution, and online training can take place to improve the model’s accuracy.

Features extracted from GSR/EDA were used in most of the studies [4,6,7,15], from PPG in [7] and from ECG (in the non-wearable setting) in [14]. In this research, we used features extracted from both PPG and EDA, temperature and pressure sensors, as well as activity-related features from the accelerometer, magnetometer, and gyroscope. Additionally, some personal features such as age, height, weight, and gender were recorded. Dehydration monitoring can benefit from other health applications that quantify the duration of activity of the person in running, cycling, walking, and other activities. Its output can also be integrated with other diet applications to ensure a healthy lifestyle for the users.

All the studies experimented with different types of models; they reported the best accuracy for different models according to the data used: random forest in [4], decision trees in [15], K-NN in [6], SE-KNN and Cubic SVM in [7], and SVM in [14]. In this research, tree-based models (extra trees and random forest) achieved the least error in prediction. Regression tree-based models and the random forest algorithm are more immune to multicollinearity by design and showed less training time in the experiments. In the case of dehydration monitoring and similar applications affected by the personal features of the subject, transfer learning needs to be performed on-device to update the model. Taking into account the computational limitations of the device, using algorithms with a short training time/low complexity could be of great benefit in dehydration monitoring from data. In terms of interpretability and explainable AI, tree-based models have also proven to achieve the best interpretability characteristics by design [31]. Interpretability is especially required by healthcare applications to provide an informed and clear justification for the model decisions. For other types of black-box models that are not interpretable, explainable AI techniques, such as features relevance and visualizations, are utilized to promote confidence in the model, fairness, and informativeness [31]. Using SHAP [32] to visualize the Shapley values for how the features impact the output of the model is shown in Figure 9.

As mentioned in the last section, optimization techniques for on-device deployment to decrease the models’ sizes are essential for the limited memory resources on wearable devices. Optimizing the training time and inference time consequently is important due to the computational resources’ limitations. Regarding the model size and training time comparisons, the other studies did not report their results in this aspect. Only [15] performed power consumption analysis on an Android platform, as mentioned in the Section 2.

Regarding the dataset size, our dataset was second in terms of the total minutes recorded after [15], but we collected data for eleven subjects compared to only five subjects for them. The Shimmer wearable device used in this study is a non-water proof device that cannot be used in a 24 h monitoring scenario, as they did. The dataset collected in this study had the advantage of being heterogeneous as the length of the data collected per subject was not equal. This resembles to some extent the wearable device scenario where subjects wear the device for different periods of time. Thus, it may be used for testing in a federated learning setting. The data collection was challenging especially during the current situation due to the spread of COVID-19 and its variants. We believe that a dataset size of 11 subjects is still a limited number, and collecting data from more subjects would be important to gain more confidence in the results. Data generation and augmentation techniques using generative adversarial networks (GANs) can be researched for this purpose to improve the generalizability. The collected dataset had another limitation of not representing possible target users that we mentioned (e.g., athletes, soldiers, workers in extreme hot conditions, elderly people, etc.). Including samples representing these target users would be of great benefit to capture the variance in the features related to movement, temperature, and health conditions.

Summing up, this study modeled the problem of hydration monitoring as a regression problem rather than classification for the reasons highlighted in the paper. It integrated sensors of multiple modalities, and new features were constructed that have not been used in previous studies. The model evaluation based on accuracy, training time, and model size for optimization for on-device deployment, as well as testing the best models in a transfer learning setting were performed. The main shortcomings are the limited dataset size and missing the representation of important target users of the application. Testing on an embedded prototype was important to evaluate power consumption and gain more confidence in the applicability of the solution.

## 7. Conclusions

In this paper, the problem of dehydration monitoring was approached as a regression problem. The integration of different sensor modalities was employed, and new features were constructed from sensors’ data, besides using some personalized features. Machine learning techniques were compared according to the error in prediction metrics, as well as the training time and model size to optimize for on-device deployment. The evaluation was based on an on-device learning scenario to guarantee privacy preservation of the subjects’ data. In conclusion, hydration monitoring and prediction of the last drinking time with medical wearable data are feasible. This can help elderly people who cannot communicate their feeling of thirst, people with adipsia, and those working in very hot or very cold weather to be alerted when they need to drink water. The results presented in this paper encourage testing with a larger dataset for more subjects with a higher diversity in different working conditions and benchmarking on an embedded wearable hardware to gain more confidence in the results. 

## Figures and Tables

**Figure 1 sensors-22-01887-f001:**
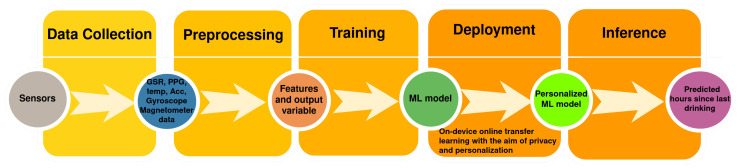
Pipeline for on-device personalized private learning for dehydration detection.

**Figure 2 sensors-22-01887-f002:**
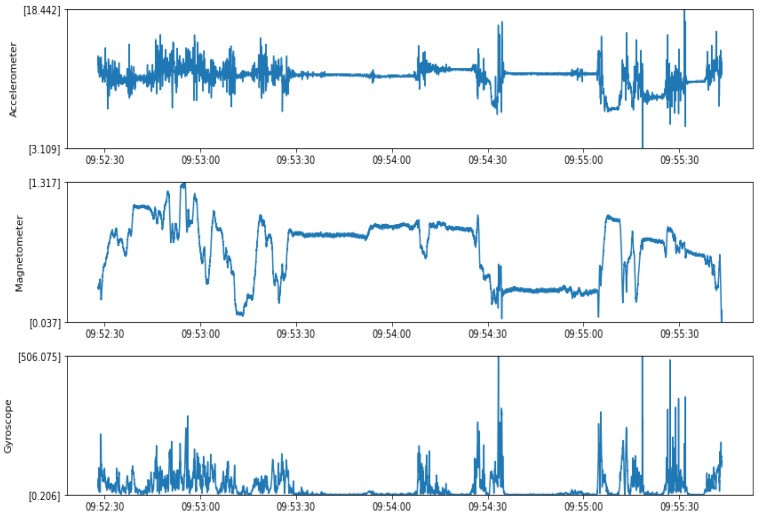
Accelerometer, magnetometer, and gyroscope sample magnitude data before preprocessing.

**Figure 3 sensors-22-01887-f003:**
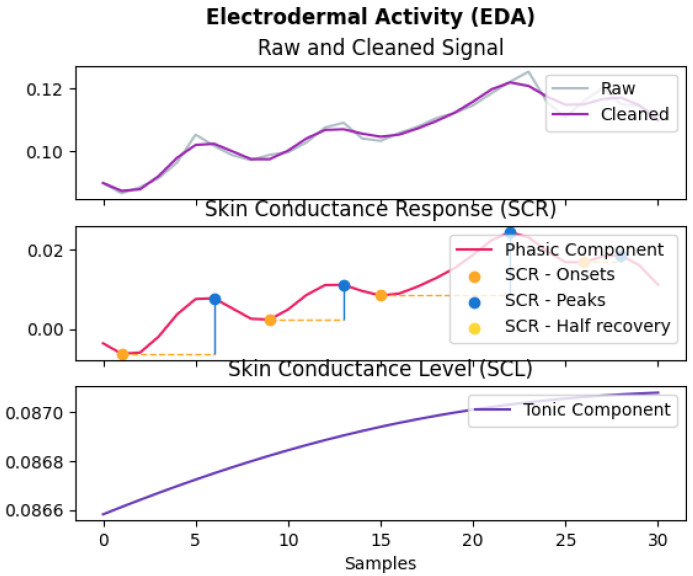
Galvanic skin response/electrodermal activity sample data.

**Figure 4 sensors-22-01887-f004:**
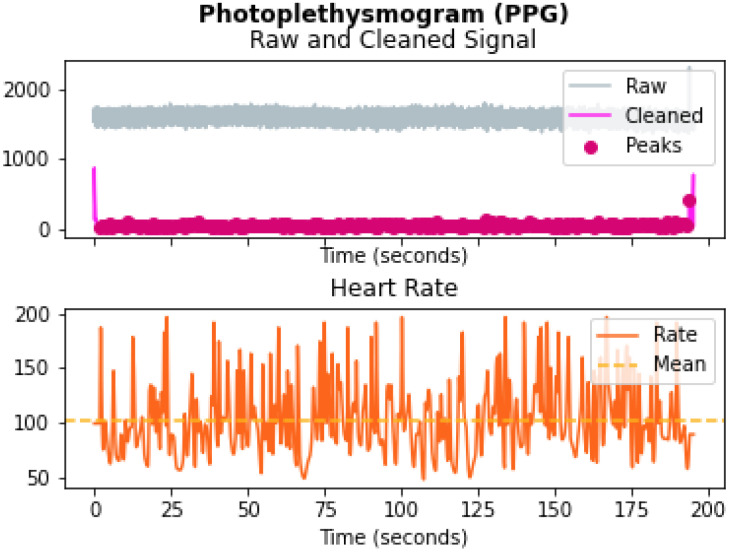
PPG sample data.

**Figure 5 sensors-22-01887-f005:**
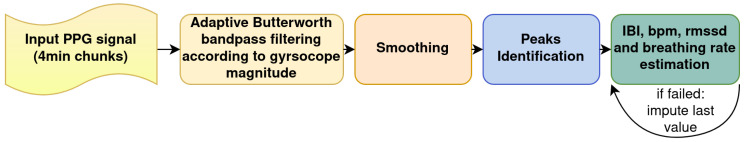
Preprocessing of the PPG signal and the extraction of features.

**Figure 6 sensors-22-01887-f006:**
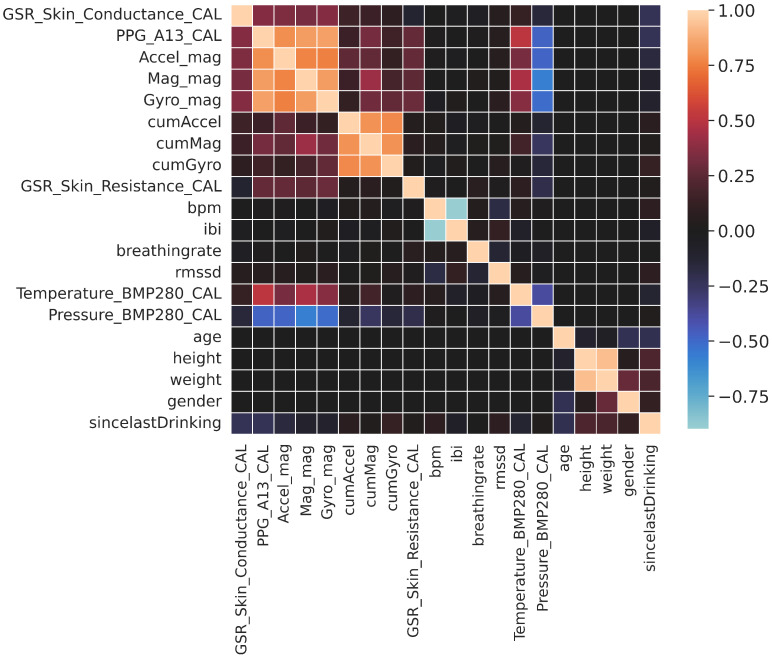
Features’ correlation matrix (please refer to Table 2 for the meaning of the features).

**Figure 7 sensors-22-01887-f007:**
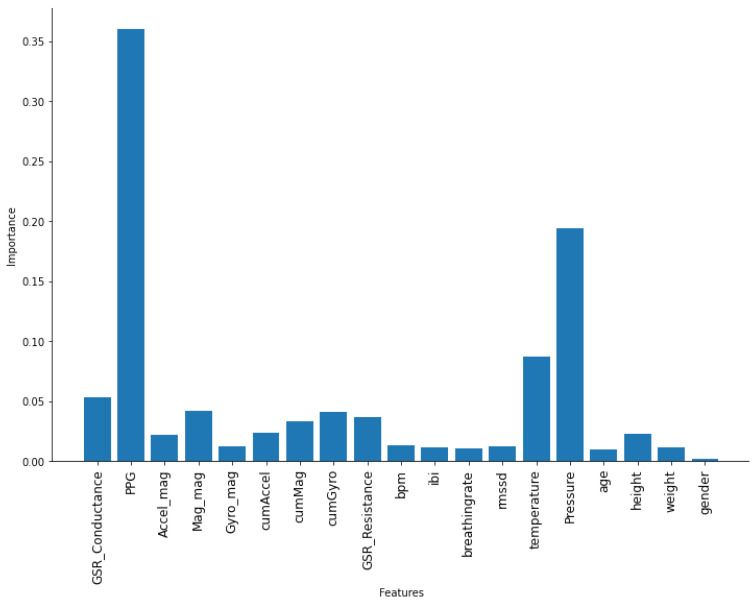
Features’ importance with random forest.

**Figure 8 sensors-22-01887-f008:**
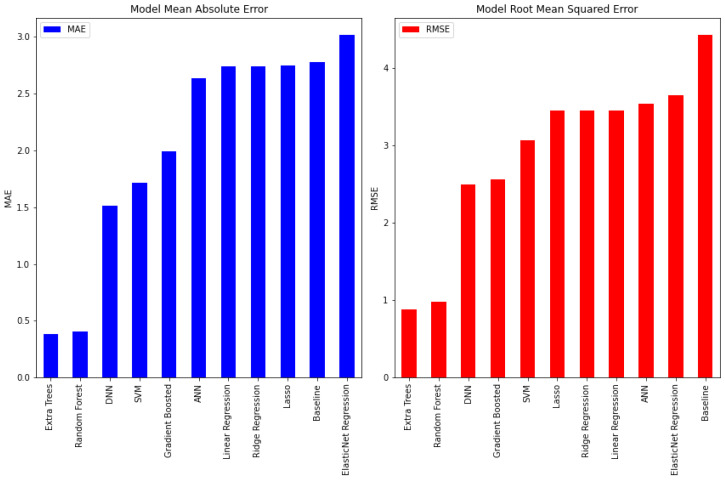
Comparison of machine learning techniques in terms of the MAE and RMSE for FEAT1.

**Figure 9 sensors-22-01887-f009:**
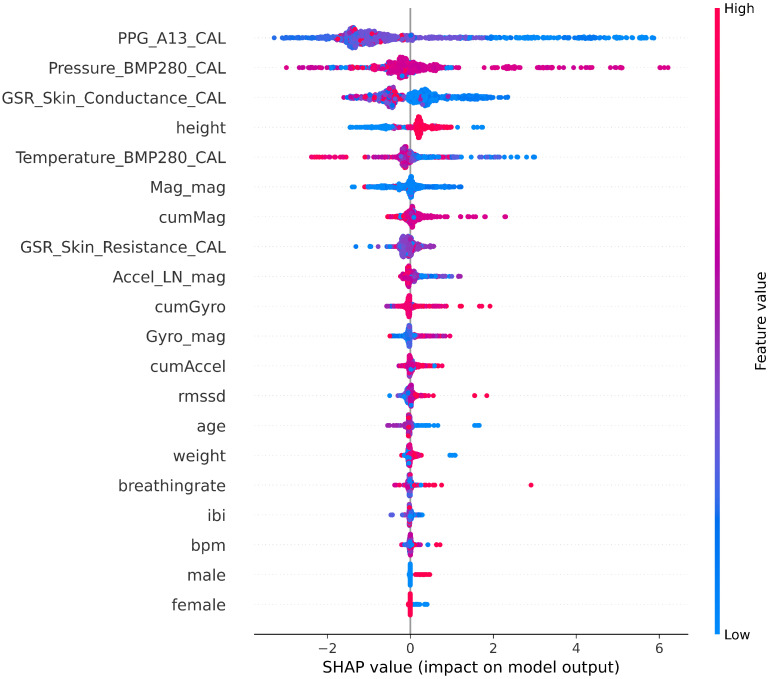
Shapley values for the random forest model using SHAP to show the impact of each feature on the model output (please refer to Table 2 for the meaning of the features).

**Table 1 sensors-22-01887-t001:** Descriptive statistics for the collected dataset.

Subject	Total Recorded Duration (min)	Maximum Fasting Duration (h)	Mean HR (bpm)	|Accelerometer| (Min., Mean, Max.) (m/s^2^)	|Magnetometer| (Min., Mean, Max.) (Ga)	|Gyroscope| (Min., Mean, Max.) (deg/s)
S1	1768	15.3	80	(5.9, 11.9, 27.7)	(0.3, 2.3, 64.7)	(1.5, 25.3, 574.2)
S2	636	13.2	67	(6.4, 10.8, 27.7)	(0.6, 2.7, 71.4)	(1.6, 26.8, 616.3)
S3	182	15	86	(8.1, 11.3, 12.1)	(0.3, 0.9, 1.6)	(2.7, 5.2, 76.9)
S4	573	12.1	83	(8, 11.6, 27.7)	(0.4, 1.8, 53.9)	(1.5, 20.1, 458.1)
S5	82	11	81	(5.8, 11.3, 12.1)	(0.4, 0.9, 1.3)	(1.7, 25.5, 78.6)
S6	23	11.2	84	(9.5, 11.4, 12.3)	(0.4, 0.6, 0.7)	(3.4, 23.9, 57.7)
S7	24	1.4	61	(10.8, 11.9, 12.3)	(0.4, 0.5, 0.8)	(1.7, 4.3, 17.7)
S8	21	1.4	81	(6.9, 8.4, 11)	(0.6, 1, 1.1)	(1.7, 13, 33.2)
S9	23	1.4	84	(10.9, 11.8, 12)	(0.4, 0.6, 0.7)	(1.6, 2.4, 9.6)
S10	22	1.4	118	(10.1, 10.9, 11.3)	(0.6, 0.9, 1)	(8.1, 20.1, 66.3)
S11	32	1.5	85	(10.5, 11.7, 12.3)	(0.4, 0.5, 0.6)	(1.7, 6.4, 27.6)

**Table 2 sensors-22-01887-t002:** List of features.

Feature	Description	Mean	VIF
PPG_A13_CAL	Raw PPG sensor calibrated values	2925	5.97
bpm	Estimated pulse rate (beats per minute)	78	5.57
ibi	Inter-beat interval (IBI) estimated from the PPG signal	772	5.53
breathingrate	Estimated breathing rate from the PPG signal	0.22	1.08
RMSSD	Root mean square of successive differences between estimated heartbeats	75.2	1.08
GSR_Skin_Resistance_CAL	Raw calibrated GSR resistance (kOhms)	9923	1.21
GSR_Skin_Conductance_CAL	Raw calibrated GSR conductance (μS)	5	1.31
Accel_mag	Magnitude of the accelerometer, Equation (Equation 1)	11.5	5.19
Mag_mag	Magnitude of the magnetometer, Equation (Equation 3)	2	7.81
Gyro_mag	Magnitude of the gyroscope, Equation (Equation 5)	22.3	7.87
cumAccel	Cumulative accelerometer change, Equation (Equation 2)	2970	6.34
cumMag	Cumulative magnetometer change, Equation (Equation 4)	3401	7.08
cumGyro	Cumulative gyroscope change, Equation (Equation 6)	46,318	6.02
Temperature_BMP280_CAL	Surrounding temperature	34.9	1.53
Pressure_BMP280_CAL	Atmospheric pressure	99.7	1.6
age	Age of the subject	30	1.14
height	Height of the subject (cm)	159	11.23
weight	Weight of the subject (kg)	63	12.11
gender	Gender of the subject (male/female)	N/A	1.8

**Table 3 sensors-22-01887-t003:** Comparison of the machine learning techniques for FEAT1.

Model	MAE	RMSE	Training Time (s)	Size (MB)
Baseline	2.78	4.43	0.00	0.00
Linear Regression	2.74	3.45	0.00	0.0007
Lasso	2.75	3.45	0.08	0.005
Ridge Regression	2.74	3.45	0.00	0.0007
ElasticNet Regression	3.02	3.64	0.00	0.0007
SVR	1.72	3.06	0.21	0.321
ANN	2.64	3.54	8.29	0.061
Gradient Boosting	1.99	2.57	0.14	0.028
**DNN**	1.51	2.50	10.52	**0.34**
**Random Forest**	0.41	0.98	**0.97**	9.58
**Extra Trees**	**0.39**	**0.88**	16.41	30.4

**Table 4 sensors-22-01887-t004:** Comparison of the machine learning techniques for FEAT2.

Model	MAE	RMSE	Training Time (s)	Size (MB)
Baseline	2.91	4.64	0.00	0.00
Linear Regression	3.08	3.78	0.00	0.00
Lasso	3.41	4.04	0.11	0.01
Ridge Regression	3.08	3.78	0.01	0.00
ElasticNet Regression	3.10	3.79	0.01	0.00
SVR	2.81	4.50	0.27	0.23
ANN	3.08	3.78	15.11	0.06
Gradient Boosting	2.14	2.74	0.09	0.03
**DNN **	1.14	1.90	18.40	**0.32**
**Random Forest**	0.36	0.84	**0.66**	9.15
**Extra Trees**	**0.27**	**0.72**	9.38	28.94

**Table 5 sensors-22-01887-t005:** Transfer learning results at different test sizes (0.3 and 0.7 of the subject’s samples).

	MAE (0.3/0.7)	RMSE (0.3/0.7)	Training Time (0.3/0.7)	Model Size (0.3/0.7)
DNN	2.41/3.12	3.29/4.39	6.17/4.8	0.31/0.31
Random Forest	0.58/0.7	0.90/1.2	**0.09/0.06**	**0.84/0.36**
Extra Trees	**0.41/0.67**	**0.61/0.91**	0.50/0.17	2.64/1.14

**Table 6 sensors-22-01887-t006:** Comparison of this study and related studies for dehydration monitoring.

Study	Problem	Features	Techniques	Dataset	Model Size	Training Time
[14]	Classification into three classes: rest before exercise, post-exercise, and after hydration	RR interval, RMSSD, and SDRR of ECG	SVM and K-means	30 min ECG (10 min for each class) for 16 athletes (total = 480 min)	No	No
[7]	Classification into hydrated/mild dehydrated	9 features from EDA and PPG	LDA, QDA, logistic regression, SVM, fine and medium Gaussian kernel, K-NN, decision trees, and ensemble of K-NNs	8 min EDA and PPG for each of 17 subjects (total = 136 min)	No	No
[6]	Classification into hydrated/dehydrated	9 statistical features from the GSR signal	Logistic regression, SVM, decision trees, K-NN, LDA, Naive Bayes	2 h of EDA signal for each of 5 subjects (total = 600 min)	No	No
[4]	Classification into hydrated/dehydrated	Combinations of 6 statistical features extracted from the GSR signal	Logistic regression, random forest, K-NN, naive Bayes, decision trees, LDA, AdaBoost classifier, and QDA	EDA signals for 5 subjects, but they did not mention the recorded time	No	No
[15]	Classification into well-hydrated, hydrated, dehydrated, and very dehydrated	12 features extracted from EDA and 2 features from the activity recognition model	Random forest, decision trees, naive Bayes, BayesNet, and multilayer perceptron	24 h for 5 d of EDA signals, as well as activity labeling for 5 subjects (total = 36,000 min)	No	No
This study	Regression for the number of hours since last drinking	FEAT1(19) and FEAT2(12) as described in Section 4	Linear, lasso, ridge, and ElasticNet regression, SVR, ANN, gradient boosting, DNN, random forest, and Extra Trees	Total of 3386 min for 11 subjects	Yes	Yes

## Data Availability

Raw data have been made publicly available at Zenodo: https://zenodo.org/record/6299964.

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
