# Peer review of "Towards On-Device Dehydration Monitoring Using Machine Learning from Wearable Device’s Data"

_sensors, 2022, doi:10.3390/s22051887_

Round 1

Reviewer 1 Report

The manuscript introduces a study on exploiting wearables sensors data through machine learning algorithms that predict the last drinking time for monitoring dehydration.

Although the work is systematic and thorough it is obscured by the fact that it reads more like a lab report rather than proposing a novel approach, novel tools, or targeted interventions. 

I would suggest emphasizing the contributions of the paper, the benefits of the approaches and the impact the study has. Please reorganize the sections as suggested in the following and make a clear statement of your contribution and needs for such an approach. Eventually, a closed-loop instantiation idea would also be useful for the reader to frame the contribution.

Per-section comments:

  • Abstract
    • What do you mean by" for athletes, battlefield soldiers, workers in extreme weather conditions and elderly people 
      who are not able to communicate their need for water."? Can you be more explicit? Additionally, from the 11 candidates, how many of each class are covered in the sample? How about subject stratification?
    • Abbreviations not introduced, avoid in abstract ("GSR sensor, PPG sensor")

    • "Feature analysis and selection, and results of applying different models are compared with respect to different metrics." This is textbook knowledge, where is the novelty and what makes the manuscript different from a lab report or a student thesis? This should shine in the abstract.
    • "Regression models based on different body signals and characteristics are promising in developing techniques to estimate the subject’s last drinking time" A hand-waving statement, without a clear statement on the performance of the approach, why, and how. Please, also make clear where is the contribution, a method for interpolation and/or extrapolation.
  • Keywords:
    • please avoid abbreviations or introduce them, e.g. EDA, PPG. 
  • Background and related work
    • This section reads like a lab report and the disconnected sections are floating. I would just maybe rewrite them as a synthetic table where you emphasize the actual components involved in the problems at hand and how existing approaches cover or not all of the points.

  • Machine Learning for dehydration monitoring
    • good overview but please make sure the context you follow, i.e. embedded inference, regression of drinking pattern to predict the last drinking time.
  • 4 Dataset
    • "The raw dataset is publicly available by contacting the corresponding author." If the data is publicly available please use Zenodo or similar platforms for sharing. If it is available per request is not publicly available.
    • " ... we calculated the accumulative change in accelerometer magnitude as in Equation 2 as a representative feature for the total change..." Can you please elaborate? How informative is this information, which just collapses the motion on the 3 axes into a single scalar? This approach risks actually missing the motion characteristics which induces the effect. Why not compute the Euler angles of the motion of the person and depending on the range of the motion support the algorithm identify predictors of high-level of effort triggering dehydration.
    • "Magnetometer data are measured from three channels for x,y,z magnetometer at 512 (maybe here is Hz?) sampling frequency with a magnitude sample shown in Figure 2."  Here, same as for acceleration, why not computing the Euler angles of the motion of the person and depending on the range of the motion support the algorithm identify predictors of high-level of effort triggering dehydration. When you anyway collapse all the 3 dimensions of the inertial measurements into a scalar does it make sense to sample so quickly? 
    • Please increase the quality of the features correlation matrix and support the explanation in the text as it is hard to follow, especially with cryptic names for features.
    • You mentioned briefly explainability, did you actually try to use SHAP or LIME methods to do that? You are indeed looking at decision trees and the aforementioned methods have good support for that.
  • Discussion
    • I would have expected to see a closed-loop system (given the alerting) rather than an exploration. I fail to grasp the novelty and the benefits of the approach, rather than an explorative study of applying machine learning on multiple features from multiple data sources.

    •  The overall study has merit and is interesting in the context of Ramadan but I would really like to have in the discussion a reiteration of the main advantages and benefits of the proposed method.

    • 11 Subjects is still a limited number, would the method be able to capture variability in larger samples?
  • Conclusion
    • "Many features from different sensors have been included besides using some personalized features. " Aside from this approach of feeding the models with a large number of features, or a ranked subset, I fail to capture the novelty, benefits and impact that such a system would actually have.
    • Please also add more references, to demonstrate that the work is framed in a field and, as I fail to get the novelty, more precisely references in biosignal acquisition and interpretation.

The manuscript needs serious revision and many aspects need to be improved for publication. I kindly ask the authors to thoroughly address the aforementioned points in order to strengthen the story.  

Author Response

Dear respectful reviewer,

We would like to kindly thank you a lot for your time reviewing our manuscript. Your comments were so valuable and reflect careful review. They helped us improve the quality of the manuscript and its readability. We appreciate your time so much. All your comments were addressed and the response and action(s) taken for each point are mentioned in the attached pdf file. Please kindly see the attachment.

Reviewer 2 Report

The manuscript presents the “Towards on-device dehydration monitoring using machine

learning from wearable device’s data”. Several suggestions would immediately improve the paper's readability and are severely lacking in the current manuscript.

  1. A lot of grammatical mistakes are found in this manuscript.
  2. The previous studies are weak. Add the recent studies.
  3. What are the possible shortcomings or drawbacks of the proposal in this paper? More detailed analyses and discussions are expected in the future version of the submission.
  4. Figure 6 is not readable. Redraw it.
  5. Reference citations are not in order. All citations should be in order.
  6. Figure 5 is not cited in the text.
  7. Add some more latest references from 2022.

Author Response

Dear respectful reviewer,

We would like to thank you a lot for your time reviewing our manuscript. Your comments were valuable to improve the readability and the quality of the manuscript. All your comments were addressed and the response and action(s) taken for each point are mentioned in the attached pdf file. Please see the attachment.

Round 2

Reviewer 1 Report

Thank you for addressing the comments in my review report. The report now is in a good shape. 

Reviewer 2 Report

The authors have revised this manuscript according to my comments. It can be accepted in the present form.